# The Transcription Factor CsgD Contributes to Engineered *Escherichia coli* Resistance by Regulating Biofilm Formation and Stress Responses

**DOI:** 10.3390/ijms241813681

**Published:** 2023-09-05

**Authors:** Cheng-Hai Yan, Fang-Hui Chen, Yu-Lu Yang, Yu-Fan Zhan, Richard A. Herman, Lu-Chan Gong, Sheng Sheng, Jun Wang

**Affiliations:** 1Jiangsu Key Laboratory of Sericultural Biology and Biotechnology, School of Biotechnology, Jiangsu University of Science and Technology, Zhenjiang 212100, China; biojustych@163.com (C.-H.Y.); 212211802102@stu.just.edu.cn (F.-H.C.); 18770390337@163.com (Y.-L.Y.); 211111901120@stu.just.edu.cn (Y.-F.Z.); hermanansah44@yahoo.com (R.A.H.); lcgong@just.edu.cn (L.-C.G.); parasitoids@163.com (S.S.); 2Key Laboratory of Silkworm and Mulberry Genetic Improvement, Ministry of Agriculture and Rural Affairs, Sericultural Research Institute, Chinese Academy of Agricultural Sciences, Zhenjiang 212100, China

**Keywords:** engineered *Escherichia coli*, transcription factor, CsgD, resistance, biofilm, stress response

## Abstract

The high cell density, immobilization and stability of biofilms are ideal characteristics for bacteria in resisting antibiotic therapy. CsgD is a transcription activating factor that regulates the synthesis of curly fimbriae and cellulose in *Escherichia coli*, thereby enhancing bacterial adhesion and promoting biofilm formation. To investigate the role of CsgD in biofilm formation and stress resistance in bacteria, the *csg*D deletion mutant Δ*csg*D was successfully constructed from the engineered strain *E. coli* BL21(DE3) using the CRISPR/Cas9 gene-editing system. The results demonstrated that the biofilm of Δ*csg*D decreased by 70.07% (*p* < 0.05). Additionally, the mobility and adhesion of Δ*csg*D were inhibited due to the decrease in curly fimbriae and extracellular polymeric substances. Furthermore, Δ*csg*D exhibited a significantly decreased resistance to acid, alkali and osmotic stress conditions (*p* < 0.05). RNA-Seq results revealed 491 differentially expressed genes between the parent strain and Δ*csg*D, with enrichment primarily observed in metabolism-related processes as well as cell membrane structure and catalytic activity categories. Moreover, CsgD influenced the expression of biofilm and stress response genes *pga*A, *mot*B, *fim*A, *fim*C, *ira*P, *omp*A, *osm*C, *suf*E and *ela*B, indicating that the CsgD participated in the resistance of *E. coli* by regulating the expression of biofilm and stress response. In brief, the transcription factor CsgD plays a key role in the stress resistance of *E. coli*, and is a potential target for treating and controlling biofilm.

## 1. Introduction

Biofilms are physical barriers between bacteria and the external environment and provide protection for bacteria from external stress and environmental changes [1]. Studies have shown that biofilms can prevent toxins, antibiotics and other harmful substances from entering cells, thus protecting the integrity and life activities of bacteria [2,3]. In medical and food fields, biofilms are almost insensitive to antibiotics and drugs, and the antibiotic resistance of bacteria in biofilms is 100–1000 times higher than that of bacteria in the planktonic state [4]. Biofilms are not only simple bacterial community, but also involve necessary physiological and phenotypic changes. It is a highly structured multicellular microbial community, which is composed of bacterial cells and an extracellular polymeric substance (EPS) matrix, which could provide niches and enhance stress resistance [5]. With an in-depth understanding of the formation mechanism, the advantages of biofilms in beneficial bacteria have been gradually discovered and have played an important role in environmental restoration, sewage treatment and biocatalysis [6].

*Escherichia coli* has been applied to bioengineering platforms because of its clear genetic background and simple genetic engineering operation. *E. coli* BL21(DE3) has a shorter cell division and excellent protein expression ability, which is suitable for large-scale expression of foreign proteins [7]. Meanwhile, the mature genetic engineering tools and technology of *E. coli* make it relatively simple and efficient to carry out genetic transformation. However, engineered *E. coli* is more vulnerable to multiple environmental stresses in application, especially in reaction conditions such as acids, alkalis, organic solvents and metal ions [8,9]. It is worth noting that biofilms play an important role in increasing the survivability of *E. coli* in stressful environments. Ahan et al. found that the biofilm structure could improve the stability of the whole cell biocatalyst of *E. coli* [10]. This conclusion was also verified by Tong et al., and the results showed that the enzyme-catalyzed conversion rate of recombinant strains with biofilm reached 78%, which was significantly higher than that of an immobilized enzyme (49%) and free enzyme (40%) [11]. However, the protective effect of biofilm also brings trouble to the elimination of pathogenic bacteria. Biofilms could block the entry of exogenous substances, reduce the direct attack of bacteriostatic agents on bacteria, and thus reduce the damage of stress factors to cells and enzymes [1]. Therefore, exploring the development and regulation process of biofilm was conducive to achieving the controllability of biofilm according to actual needs, including promoting or inhibiting the formation of biofilms.

The formation and regulation of *E. coli* biofilms involve a series of gene expression regulations. Among them, CsgD as an important regulatory factor is related to many physiological processes, such as extracellular polymer formation, toxin production and cell adhesion [12,13]. In the process of biofilm formation, CsgD can activate the synthesis genes of extracellular polysaccharides *csg*B and *csg*A and the synthesis genes of extracellular cellulose *csg*C and *csg*E [14]. Previous studies have shown that in the initial stage of colony formation, CsgD could regulate the expression of the Curli protein and fixes bacteria on the surface of the carrier, forming a stable population [12,15]. In addition, CsgD could also play a protective role against oxidative stress in *E. coli*. Specifically, tolerance could be enhanced by CsgD activating the expression of several genes related to oxidative stress (*kat*G, *ira*P, etc.), thus increasing the expression of antioxidant enzymes [16,17]. In general, CsgD plays an important role in the growth, metabolism and stress resistance of *E. coli*, and is a potential target for the treatment and control of *E. coli*.

In this study, *E. coli* BL21(DE3) was used as the model strain to explore the response mechanism of the transcription factor CsgD to biofilm formation and environmental stress, and the *csg*D deletion mutant strain was constructed by using the CRISPR/Cas9 gene-editing system. Firstly, the growth curve, biofilm formation ability and motility of bacteria were investigated. Then, by comparing the survival rates of the parent strains and *csg*D-deficient strains in stress environments (acid stress, alkaline stress, oxidation stress and osmosic stress), the influence of the *csg*D gene on the stress resistance of *E. coli* was evaluated. Subsequently, the differences in gene expression between the two strains were compared by transcriptome sequencing technology to further understand the regulatory mechanism of CsgD on biofilm formation in *E. coli*. In summary, revealing the important role of CsgD in the stress resistance of *E. coli* provides potential targets for the treatment and control of *E. coli*.

## 2. Results and Discussion

### 2.1. Deletion of the csgD Gene Downregulated E. coli Biofilm Formation

The *csg*D gene was knocked out by the CRISPR/Cas9 tool. As shown in Figure 1a, the sgRNA targeting the *csg*D gene was integrated into pTargetF. Then, the plasmids pTargetF-sg-*csg*D and pCas were simultaneously transformed into the parent strain *E. coli* BL21(DE3), and the *csg*D gene was accurately knocked out. Figure 1b shows the amplified fragments of the L. arm-F and R. arm-R primers in the parent strain and the *csg*D-deficient strain Δ*csg*D. The results show that the amplified fragments of the parent strains were 2051 bp, while those of Δ*csg*D were only 1400 bp. In addition, the amplified fragment was verified by sequencing, and it was confirmed that the *csg*D-deficient strain was successfully constructed.

As shown in Figure 1c, the growth curve of Δ*csg*D was not significantly different from that of the parent strain (*p* > 0.05), indicating that the *csg*D did not affect the growth of *E. coli*. The determination results of the biofilm (Figure 1d) show that the OD_595nm_ value of Δ*csg*D was only 0.66 ± 0.02, which was significantly lower than that of the parent strain (*p* < 0.05). Studies have shown that the *csg*D gene is the key regulator of biofilm formation in *E. coli*, and it regulates biofilm formation by regulating the production of the Curli protein [12]. In addition, the *csg*D gene was involved in regulating the expression of genes related to biofilm formation, such as *dgcC* and *bcs*A. Among them, cellulose synthase encoded by the *dgc*C gene could catalyze two molecules of guanosine triphosphate (GTP) to produce c-di-GMP, and the increase in c-di-GMP concentration could promote the adhesion of bacteria and the formation of biofilms [18,19]. The *bcs*A gene encodes bacterial cellulose synthase, which can synthesize membrane elements such as cellulose and increase the adhesion and aggregation of bacteria [20]. These results indicate that the deletion of the *csg*D gene exerts a negative role in *E. coli* biofilm formation.

Figure 1e,f show the EPS composition of Δ*csg*D. The results show that the protein contents in loosely bound EPS (LB-EPS) and tightly bound EPS (TB-EPS) of Δ*csg*D were 17.00 ± 7.94 mg/L and 35.60 ± 2.00 mg/L, respectively, which were significantly reduced by 88.90% and 55.80% compared with the parent strains (*p* < 0.05). Meanwhile, the polysaccharides in the LB-EPS and TB-EPS of Δ*csg*D were significantly reduced by 17.80% and 62.90%, respectively, compared with the parent strains (*p* < 0.05). Extracellular proteins and polysaccharides are the main components of biofilms, which provide the necessary framework for biofilm formation [21]. In the initial stage of biofilm formation, extracellular polysaccharides could help bacteria firmly adsorb on the surface of objects and stimulate bacterial aggregation [21,22]. Therefore, the abnormal expression of the *csg*D gene would inhibit the synthesis of extracellular polymer substances, inhibit the adhesion of bacteria and the formation of a fibrous web, and then affect the formation of biofilms and their adaptability to the environment.

### 2.2. Deletion of the csgD Gene Influences the E. coli Morphotype

Figure 2a shows the biofilm external morphology of Δ*csg*D by scanning electron microscopy (SEM). The morphologies of the two strains were complete, and the rod-like structure of *E. coli* was evident. The parent strains adhere closely, and the EPS could help bacteria stack together, forming a three-dimensional biofilm with a certain thickness [14]. However, the biofilm of Δ*csg*D was loose, and the adhesion strength was reduced. The results of crystal violet staining showed consistent results (Appendix A). As shown in Figure 2b and Appendix A, the structure of the biofilm was observed more intuitively by laser confocal microscopy and fluorescence inversion microscopy. The results showed that the fluorescence intensity of the parent strain (7.01 AU) was significantly higher than that of Δ*csg*D (2.15 AU), and thick biofilms covering the whole surface were formed. In contrast, the cell aggregation property of Δ*csg*D was decreased, and the biofilm thickness was also reduced. The above results further indicate that the deletion of the *csg*D gene has a significant inhibitory effect on the formation of *E. coli* biofilms.

### 2.3. Motility and Environmental Stress Resistance of the csgD Gene Deletion Strain

Figure 3a–c show the influence of *csg*D gene deletion on the motility of the strain. The results show that the radii of swimming, twitching and swarming motility of Δ*csg*D were 0.38 ± 0.02 cm, 0.54 ± 0.01 cm and 0.44 ± 0.04 cm, which were 33.30%, 22.80% and 20.00% lower than those of the parent strain, respectively (*p* < 0.05). Previous studies have found that CsgD has a complex influence on the motility of *E. coli* [23]. CsgD participates in the basic physiological processes of bacteria, such as cell division and metabolic regulation, which are closely related to cell movement [24]. On the other hand, CsgD is also involved in regulating the formation and movement of bacterial flagella [25,26]. Therefore, the deletion of the *csg*D gene decreased the comprehensive motility of *E. coli* and weakened physiological processes such as spreading.

Figure 3d,e show the antioxidant properties of the parent strain and *csg*D-deficient strain. The results show that the reactive oxygen species (ROS) accumulation of Δ*csg*D was increased by 31.19%, while the catalase activity was decreased by 50.14%. CsgD was shown to have a stabilizing effect on σS protein levels, thus protecting cells from oxidative damage [27]. The sensitivity of *E. coli* with *csg*D gene deletion to oxidative stress would affect the balance between ROS and antioxidant enzymes [28]. In short, it would lead to an increase in intracellular reactive oxygen species, resulting in the weakening or even loss of the oxidative protection mechanism [16,29]. Therefore, the absence of the *csg*D gene leads to the oxidative stress reaction of cells and a decline in resistance to the external environment.

Figure 3f–i show that the survival rate of Δ*csg*D was significantly lower than that of the parent strain under acid, alkali and osmotic pressure (*p* < 0.05). Under oxidative stress, there was no significant difference in the survival rate between the two strains (*p* > 0.05) because the original strain *E. coli* BL21(DE3) had weak tolerance and defense mechanisms against oxidative stress [9,30]. Although the survival rate of Δ*csg*D in a hydrogen peroxide environment was only 2.60 ± 0.14 × 10^5^ CFU/mL, there was no significant difference from the parent strain (*p* > 0.05). Previous research has shown that in stressful environments, such as acid, alkali and osmotic pressure, the transcription level of *csg*D increases and promotes the synthesis of extracellular aggregates and cellulose, thereby helping bacteria respond to stimuli [31,32]. The above results indicate that CsgD might help *E. coli* adapt to the extracellular environment and survive under stress conditions.

### 2.4. Transcriptome Analysis of the csgD-Deficient Mutant Strain

To systematically explore the influence of CsgD on the biological process of *E. coli* in coping with environmental stress, the changes in gene transcription level between the parent strain *E. coli* BL21(DE3) and the mutant Δ*csg*D were compared by RNA-Seq. Six cDNA libraries were constructed, which were three biological repeats of *E. coli* BL21(DE3) and the mutant Δ*csg*D. As shown in Table 1, the GC contents of the six libraries were all approximately 50%, and all Q30% values were equal to or greater than 93.61%. By screening high-quality sequences, a total of 158,587,042 reads were obtained. The results indicated that the sequencing data were qualified and could be further analyzed.

CsgD is involved in many biological processes of *E. coli*, including bacterial biofilm formation, colony formation, antibiotic resistance and stress response [14]. The differentially expressed genes in the two strains were screened with the criteria of log 2 (fold change) ≥ 1 and *p* ≤ 0.05 [30], and the results are shown in Figure 4a,b. The results show that the expression of 491 genes (accounting for 11.55% of the total genes) was significantly different between the parent strains and the mutant Δ*csg*D (*p* < 0.05), among which 176 genes were upregulated and 315 genes were downregulated by CsgD. These results indicate that CsgD exerts a global influence on the gene expression of *E. coli*.

To further explore whether CsgD might be involved in the regulatory pathways of *E. coli*, KEGG functional enrichment analysis was carried out on the screened differentially expressed genes. As shown in Figure 4c, the differentially expressed genes were mainly enriched in ABC transporters, the two-component system, galactose metabolism, biofilm formation and pyrimidine metabolism. Figure 4d shows the GO function enrichment analysis of the differentially expressed genes. According to gene function annotation, 491 differentially expressed genes were annotated into three GO categories, and further enriched in secondary functional nodes [30]. According to the annotation, differentially expressed genes were mainly enriched in the cellular process and metabolic process in biological process, the cellular anatomical entity and intracellular in cellular components, and the catalytic action and binding in molecular functions. It is worth noting that all the genes involved in biological adhesion were downregulated, indicating that the deletion of *csg*D gene significantly inhibits bacterial adhesion. Specifically, CsgD regulates the expression of several genes, including *csg*B, *csg*A and *agf*D, which encode proteins for the synthesis of collagen fibronectin [12,13]. When the expression level of CsgD increases, the biosynthesis of collagen fibronectin increases, thus enhancing the adhesion of cells [18]. Therefore, the differentially expressed genes caused by *csg*D deletion in *E. coli* were mainly involved in metabolism, the cell membrane and catalytic activity, which indicated that CsgD could directly or indirectly regulate diverse metabolism-related genes.

### 2.5. Validation of Transcriptome Analysis

Figure 5 shows the correlation between the transcriptome data and gene expression levels obtained by RT-qPCR. To validate the reliability of the transcriptomic data, eight differentially expressed genes were randomly selected for analysis of their relative expression levels using RT-qPCR [33]. Among them, half of the genes showed upregulation and the other half showed downregulation. As shown in Figure 5a, the differential expression trends of these genes were consistent with the transcriptomic data. Linear regression was performed on the correlation between RT-qPCR and RNA-seq data (Figure 5b), in which R^2^ was 0.9603 and the slope was 1.7165, indicating that RT-qPCR data were positively correlated with the transcriptome data.

### 2.6. Deletion of the csgD Gene Downregulated the Expression of Biofilms and Stress Response Genes

To further explore the biological function of CsgD in coping with environmental stress, the cell adhesion-related genes (*pga*A, *mot*B, *fim*A and *fim*C) and response to stimulus-related genes (*ira*P, *omp*A, *osm*C, *suf*E and *ela*B) of the parent strains and *csg*D deletion strains were detected by RT-qPCR. The functional annotations of these differential genes in RNA-seq were shown in Table 2. As shown in Figure 6, the relative expression levels of the *pga*A, *mot*B, *fim*A and *fim*C genes in *csg*D-deficient strains were 0.21, 0.59, 0.19 and 0.01, respectively (*p* < 0.05), indicating that the adhesion ability of bacteria was significantly inhibited. The protein encoded by the *pga*A gene could promote the synthesis and assembly of poly-*β*-1,6-N-acetyl-D-glucosamine (PGA) and participate in the immune escape, adhesion invasion and biofilm formation of bacteria [34,35]. Therefore, the absence of *csg*D inhibits the synthesis of PGA and the adhesion process of bacteria. The MotB protein is an integral part of bacterial motility, which participates in bacterial swimming and chemotaxis and enables bacteria to swim effectively and escape stress in the environment [36]. The genes *fim*A and *fim*C regulate the synthesis and assembly of fimbria, respectively, which play an important role in the process of bacterial adhesion and biofilm formation [14,37]. Therefore, there was positive feedback regulation between *csg*D and the genes related to bacterial adhesion. The deletion of the *csg*D gene inhibits the synthesis of flagella and fimbriae in *E. coli* and weakens the adhesion ability of the biofilm and cells.

As shown in Figure 6, the relative expression levels of *ira*P, *omp*A, *osm*C, *suf*E and *ela*B in the *csg*D-deficient strains were 0.48, 0.36, 0.40, 0.73 and 0.20 respectively, which were significantly lower than those of the parent strain (*p* < 0.05). Among them, *ira*P could affect the stress response and gene expression regulation of bacteria by regulating the stability of the RpoS protein, participating in the oxidative stress response [17,38]. The protein encoded by *omp*A can bind to the receptor on the surface of the host cell so that bacteria can adhere to the host cell closely, thus achieving bacterial invasion [39]. In addition, *omp*A could enhance the stability of the bacterial outer membrane and help bacteria resist the invasion of environmental factors such as external mechanical pressure [40]. When bacteria are faced with oxidative stress, reactive oxygen species cause damage to proteins, DNA and other molecules in cells. The antioxidant enzymes encoded by *osm*C can play the role of antioxidant defense, eliminating harmful substances and participating in the stress response in bacteria [41]. The SufE protein provides an antioxidant defense mechanism by participating in the synthesis and assembly of ferritin and protects cells from oxidative stress [42]. Endomembrane proteins are encoded by *ela*B and participate in various stress reactions. Studies have shown that the absence of *ela*B reduces the survivability of cells under heat stress and oxidative stress [43]. Compared with the parent strains, the transcription levels of these genes were significantly reduced (*p* < 0.05). Therefore, CsgD has a positive regulatory effect on the expression of stress response genes, and its abnormal expression would reduce the response of *E. coli* to stimulation.

## 3. Materials and Methods

### 3.1. Bacterial Strains, Plasmids and Growth Conditions

The *E. coli* BL21(DE3) was obtained from Beijing TransGen Biotech Co., Ltd. (Beijing, China). All strains, with a concentration of 10^7^ CFU/mL, were inoculated at a 1% (*v*/*v*) ratio in a Luria–Bertani (LB) medium. The strains were cultured at 37 °C for 12–14 h until the optical density at 600 nm (OD600) reached ~0.6, and the bacterial density was determined by gradient dilution method. Kanamycin (50 μg/mL) and spectinomycin (50 μg/mL) were added if needed. A complete list of all the strains and plasmids used in this study can be found in Appendix A.

### 3.2. Construction of the csgD Deletion Strain

The *csg*D-deficient strain was constructed from the parent strain *E. coli* BL21(DE3) by the CRISPR/Cas9 system and named Δ*csg*D [44]. The primers used are shown in Appendix A. Briefly, the sgRNA of the *csg*D gene was designed through the website https://www.zlab.bio/guide-design-resources (accessed on 31 March 2023), and the designed sgRNA sequence was added to the primer. Plasmid pTarget and sgRNA were assembled by whole-plasmid PCR technology, and the constructed plasmid was named pTarget-sg-*csg*D. The genomic DNA of *E. coli* BL21(DE3) was used to amplify the *csg*D upstream and downstream fragments using L. arm-F/R and R. arm-F/R primers, respectively. The upstream and downstream fragments were connected by the overlap technique to obtain homologous repair donor DNA. The plasmid pTarget-sg-*csg*D and donor DNA were electroporated into competent *E. coli* BL21(DE3) cells containing the pCas plasmid, and the *csg*D gene was deleted. Finally, the mutant strains were identified using L. arm-F and R. arm-R primers.

The knockout plasmids pTarget-sg-*csg*D and pCas were eliminated as described [44]. In short, the mutant strain identified correctly above was inoculated in an LB medium containing kanamycin and isopropyl-beta-D-thiogalactopyranoside (0.5 mmol/L). After culturing for 8–16 h, the cells were coated on an LB solid medium containing kanamycin (50 μg/mL). Then, the sensitivity of the monoclonal strain to spectinomycin (50 μg/mL) was determined, and it was verified that the pTarget-sg-*csg*D had been eliminated. Finally, the cloned strain was cultured at 43 °C overnight to eliminate the pCas plasmid.

### 3.3. Growth Curves and Biofilm Formation Assay

The growth curves of the *E. coli* BL21(DE3) and Δ*csg*D strains were monitored. The seed liquid of the two strains with a concentration of 10^7^ CFU/mL was inoculated in a liquid LB medium at a volume ratio of 1:100. And the LB medium of uninoculated strains served as a negative control. Subsequently, the strains were cultured on a shaking table at 37 °C and 180 rpm. The cell density was detected every 2 h by a multimode microtiter plate reader (Spectra Max i3, Sunnyvale, CA, USA), and a growth curve was plotted.

The biofilm biomass of *E. coli* was determined using crystal violet as reported previously [5,45]. The seed liquid of the two strains with a concentration of 10^7^ CFU/mL was inoculated into a 96-well plate, and it was cultured at 37 °C for 24 h. Each group was set with 6 wells as parallel, and the wells without inoculated strains were regarded as a negative control. After culturing, the supernatant was discarded, and the planktonic bacteria were washed with a PBS buffer solution. After natural drying, the biofilms were dyed with a crystal violet solution at a concentration of 0.1% for 30 min, and then gently rinsed with a PBS buffer solution three times. Finally, the crystal violet attached to the biofilm was dissolved using 95% ethanol, and OD_595nm_ was measured by a multimode microtiter plate reader to determine the biofilm biomass.

### 3.4. Extracellular Polymeric Substance Analysis

The EPS in the biofilms were extracted by thermal extraction [46]. The cultured biofilm was collected and quickly resuspended in 0.05% NaCl solution at 70 °C, and the supernatant was centrifuged to obtain LB-EPS. The remaining precipitate was resuspended in 0.05% NaCl solution and bathed in water at 70 °C for 30 min, and the supernatant was centrifuged to obtain TB-EPS. The protein and polysaccharide contents in the EPS were quantified by Coomassie Brilliant Blue G-250 and the phenol–sulfuric acid method [47].

### 3.5. Motility Analysis of Mutant Strain

The motility of the *E. coli* BL21(DE3) and Δ*csg*D was analyzed with slight modifications [48]. The seed liquid of the two strains was inoculated in an LB semisolid or solid medium containing 0.3%, 0.5% and 1.15% agar for swimming motility, swarming motility and twitching motility. After culturing at 37 °C for 48 h, the diameter of the colony movement area was determined.

### 3.6. Environmental Tolerance Analysis

To measure ROS, the strains were mixed with 1‰ 2′,7′-dichlorodihydrofluorescein diacetate (DCFH-DA) solution and incubated in the dark for 20 min. After incubation, the supernatant was removed by centrifugation and washed with deionized water three times. After resuspension in deionized water, the green fluorescence signal was measured by a multimode microtiter plate reader [49]. Specifically, catalase (CAT) activity was determined by spectrometry assays using the CAT assay kit following the manufacturer’s instructions.

Stress resistance to different environments was determined as described [30]. The *E. coli* BL21(DE3) and Δ*csg*D strains were cultured at 37 °C for 12 h, centrifuged at 10,000 rpm and resuspended in a PBS buffer of equal volume. Then, the resuspended bacterial liquid was mixed with an LB medium (pH 3), Tris-HCl buffer (pH 10), hydrogen peroxide (10 mM) and NaCl solution (4.8 M) at volume ratios of 1:9, 1:9, 1:1 and 1:1 and incubated at 37 °C for 1 h. Viable bacteria were counted on agar plates by the gradient dilution method.

### 3.7. Colony Morphology and Biofilm Characterization

Sterile glass slides were placed in a 12-well plate, inoculated with 1% bacterial seed solution with a concentration of 10^7^ CFU/mL and placed in an incubator at 37 °C for 24 h. After culturing, the floating bacteria were gently washed off with a PBS buffer, which was repeated three times. Subsequently, 4% glutaraldehyde was added to a 12-well plate and placed in a refrigerator at 4 °C overnight. Then, the 4% glutaraldehyde was removed, and the biofilm adsorbed on the glass slide was stained with SYBR Green I dye for 30 min, which was used for fluorescence inversion microscopy (Eclipse TS100, Nikon, Tokyo Metropolis, Japan) and laser focusing scanning microscopy (LEXTOLS4000, Olympus, Tokyo, Japan) observation according to the method of Chen and Zhu et al. [5,50]. In addition, ethanol was used to dehydrate the biofilm fixed with 4% glutaraldehyde and freeze-dried for scanning electron microscope (Regulus-8100, Hitachi, Tokyo, Japan) observation [51].

### 3.8. RNA-Sequencing Analysis

The *E. coli* BL21(DE3) and Δ*csg*D strains were cultured overnight, and the bacterial liquid was centrifuged at 8000 rpm for 10 min. The supernatant was removed, and the sample was frozen in liquid nitrogen. Three biological replicates were prepared for each treatment. The obtained samples were sent to Shanghai Biozeron Biotechnology Co., Ltd. (Shanghai, China) for RNA-seq analysis. The detailed description of the sequencing process is shown in the Appendix A. Meanwhile, the screening criteria for differentially expressed genes were log 2 (fold change) ≥ 1 and *p* ≤ 0.05 [52]. RNA-seq data was uploaded to the National Center for Biotechnology Information gene expression database (https://www.ncbi.nlm.nih.gov (accessed on 25 July 2023)) with Sequence Read Archive (SRA) accession number PRJNA998482.

### 3.9. RNA Isolation and RT-qPCR

A total RNA extraction kit (TaKaRa, Osaka, Japan) was used to extract RNA from the *E. coli*. Subsequently, cDNA was synthesized with the Hifair^®^ AdvanceFast 1st Strand cDNA Synthesis Kit. Real-time quantitative PCR (RT-qPCR) was performed with a Hieff^®^ qPCR SYBR Green Master Mix. The reference gene was *dna*E, and the designed primers are shown in Appendix A. The Ct values of samples were read in the LightCycler^®^ 96 RealTime PCR system (Roche, Basel, Switzerland), and the gene expression was analyzed by the 2^−ΔΔCt^ method [53].

### 3.10. Statistical Analysis

In the statistical analysis of data, three repeated experiments were designed to ensure the reliability and accuracy of the results. The Pearson correlation coefficient was used to determine the significance between the data, and *p* < 0.05 was considered to be significant.

## 4. Conclusions

We have proved that the transcription factor CsgD was essential for the biological characteristics of engineered *E. coli*, including biofilm formation, motility, stress response and so on. Transcriptome results show that the CsgD could directly or indirectly regulate metabolism, cell membrane and catalytic activity. More importantly, CsgD has a positive regulatory effect on genes related to bacterial flagella synthesis, adhesion and stress response, and its abnormal expression would reduce the environmental stress resistance of engineered *E. coli*. In conclusion, revealing the molecular mechanism of CsgD in the stress resistance provides potential targets for the treatment and control of *E. coli*.

## Figures and Tables

**Figure 1 ijms-24-13681-f001:**
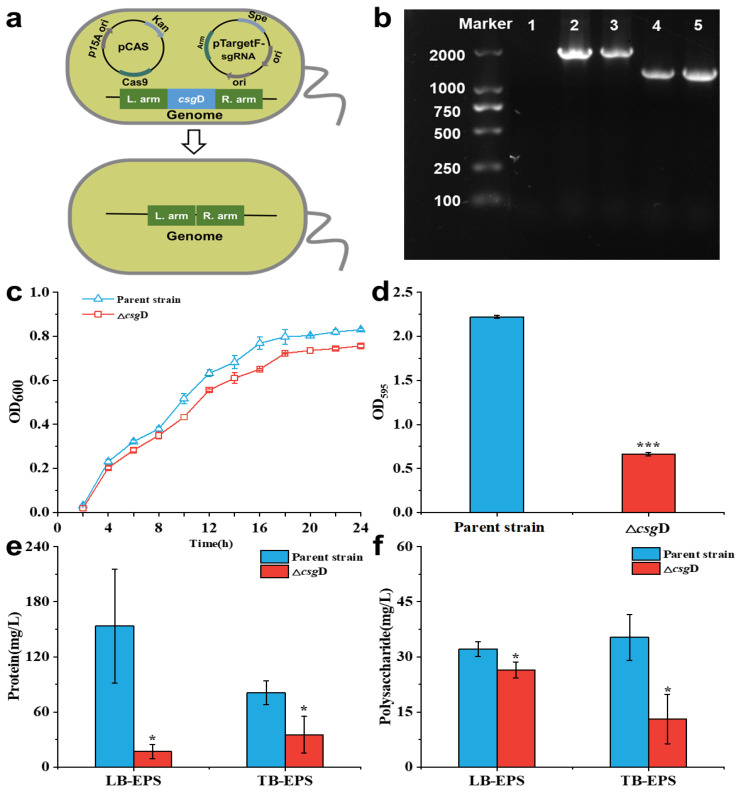
Identification and characterization of *csg*D gene deletion strains. (**a**) Schematic diagram of the strategy for deleting the *csg*D gene in *E. coli* BL21(DE3); (**b**) Confirmation of the mutant strain Δ*csg*D. M: 2000 DNA marker; Lane 1: Blank group without adding bacteria; Lane 2–3: PCR product (2051 bp) amplified from the parent strain with primers L. arm-F/R. arm-R; Lane 4–5: PCR product (1400 bp) amplified from the mutant strain Δ*csg*D with primers L. arm-F/R. arm-R; (**c**) Growth curves of the parent strain and *csg*D gene deletion strains in LB medium at 37 °C, with the OD_600nm_ measured every 2 h; (**d**) Analysis of the biofilm ability of the parent strain and *csg*D gene deletion strains; (**e**) Variations in protein content in the tightly bound EPS (LB-EPS) and tightly bound EPS (TB-EPS) of the biofilm in the parent strains and *csg*D gene deletion strains; the protein content was determined by Coomassie Brilliant Blue G-250; (**f**) Variations in polysaccharide content in the LB-EPS and TB-EPS of the biofilm in the parent strains and *csg*D gene deletion strains; the polysaccharide content was determined by the phenol–sulfuric acid method. The asterisks indicate an observation with a statistically significant difference between the *E. coli* BL21(DE3) and Δ*csg*D strains (* *p* < 0.05, and *** *p* < 0.001).

**Figure 2 ijms-24-13681-f002:**
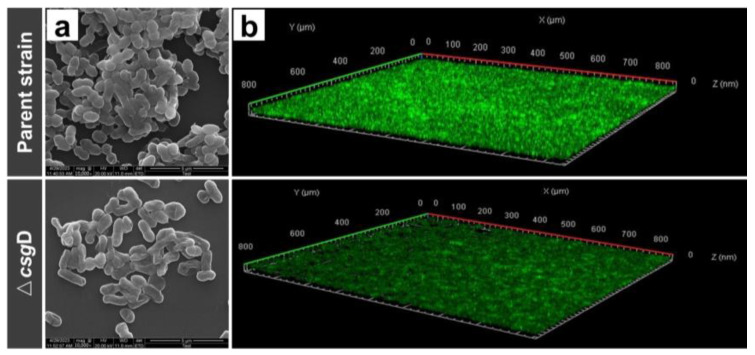
The ability of the parent strain *E. coli* BL21(DE3) and *csg*D gene deletion strains to form biofilms was tested by scanning electron microscopy (SEM) assays and laser confocal microscopy. The biofilms were stained with SYBR Green I dye. (**a**) The bacteria were cultured on a slide and incubated for 24 h. The biofilm was collected for scanning electron microscope (10,000×); and (**b**) laser confocal microscopy observations.

**Figure 3 ijms-24-13681-f003:**
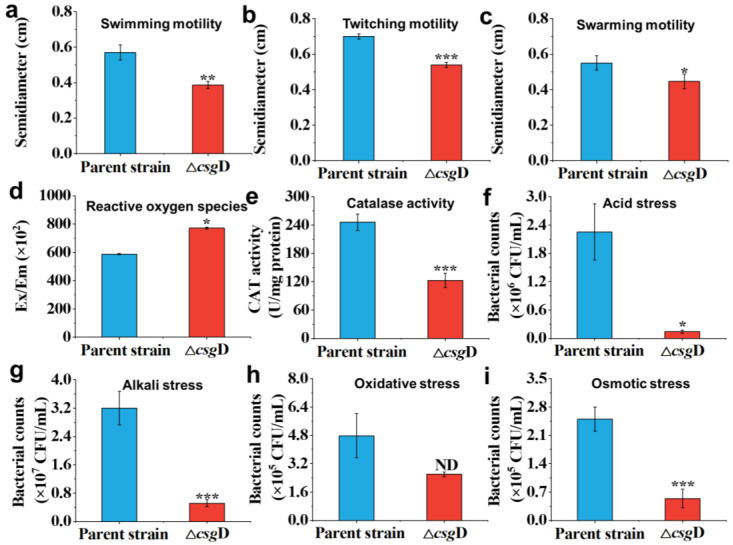
The motility and resistance to environmental stress of the parent strain *E. coli* BL21(DE3) and *csg*D gene deletion strains. The swimming (**a**), twitching (**b**) and swarming (**c**) motility of strains were measured in an LB semisolid medium containing 0.3%, 0.5% and 1.15% agar. The antioxidant capacity and survival ability of the strains were tested under diverse environmental stresses. (**d**) The content of reactive oxygen species (ROS); (**e**) catalase activity; (**f**) acid stress (pH 3); (**g**) alkali stress (pH 10); (**h**) oxidative stress (10 mM H_2_O_2_); (**i**) osmotic stress (4.8 mol/L NaCl). The asterisks indicate an observation with a statistically significant difference between the *E. coli* BL21(DE3) and Δ*csg*D strains (* *p* < 0.05, ** *p* < 0.01, and *** *p* < 0.001).

**Figure 4 ijms-24-13681-f004:**
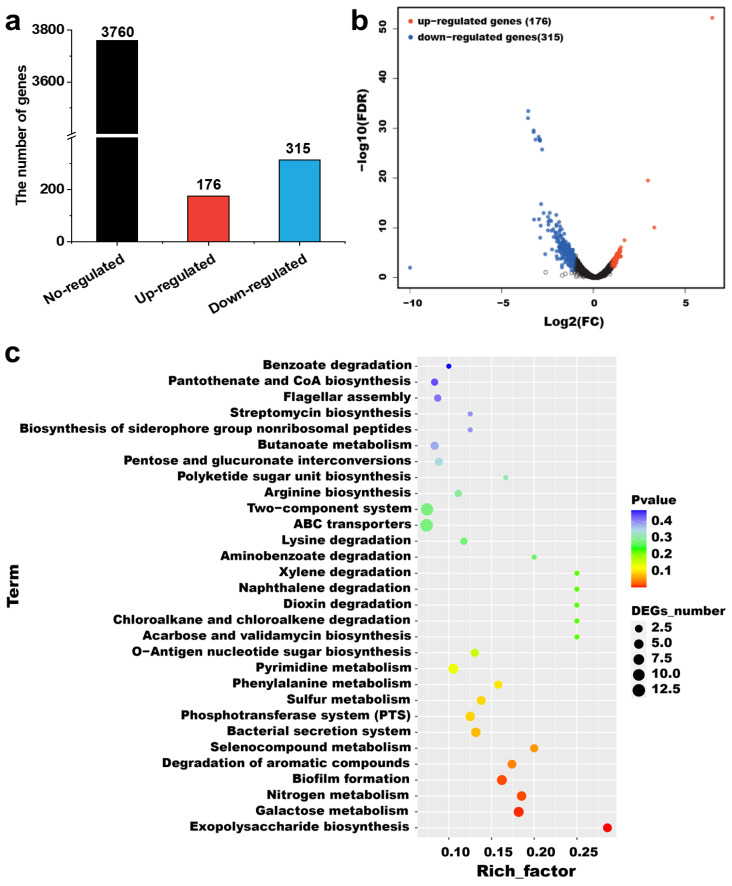
Expression of differentially expressed genes between the parent strain *E. coli* BL21(DE3) and *csg*D gene deletion strains. (**a**) Histogram illustrating the number of differentially expressed genes and nonregulated genes between the parent strain and Δ*csg*D strain; (**b**) Scatter plot of coexpressed genes between the parent strain and Δ*csg*D strain. Red, blue and black denote upregulated, downregulated and nonregulated genes, respectively, in Δ*csg*D compared with the parent strain based on the following criteria: log2 (fold change) ≥ 1 and adjusted *p* < 0.05. KEGG (**c**) and GO (**d**) pathway enrichment analyses of differentially expressed genes. The enrichment factor represents the ratio of differentially expressed genes annotated in this pathway term to all differentially expressed gene numbers annotated with this pathway term. A higher enrichment factor indicates a greater degree of pathway enrichment.

**Figure 5 ijms-24-13681-f005:**
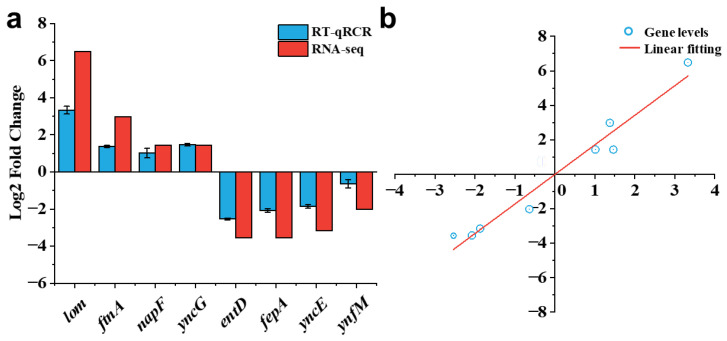
Correlation analysis between gene expression levels obtained from transcriptome data and RT-qPCR. (**a**) Expression levels (log2-fold change) obtained from RT-qPCR and transcriptome data. (**b**) Lineage analysis between the RT-qPCR and transcriptome data. The ratios obtained by RT-qPCR data (*x*-axis) were plotted against the ratios obtained by RNA-Seq data (*y*-axis), and the fitted linear equation is y = 1.7165x − 0.0118, R^2^ = 0.9603.

**Figure 6 ijms-24-13681-f006:**
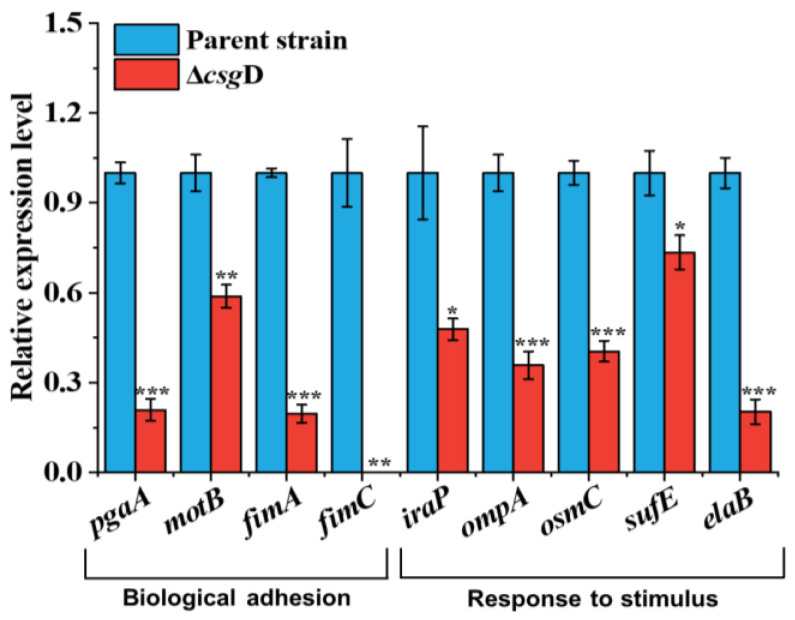
Relative expression levels of genes with biological adhesion and response to stimulus in the *E. coli* BL21(DE3) and Δ*csg*D strains were tested with RT-qPCR. Genes with biological adhesion include *pga*A, *mot*B, *fim*A and *fim*C, and genes with response to stimulus include *ira*P, *omp*A, *osm*C, *suf*E and *ela*B. Data were normalized to the housekeeping gene *dna*E. The *X*-axis represents different genes. The relative gene expression levels were calculated using the 2^−ΔΔCt^ method, and the asterisks indicate observations with a statistically significant difference between the *E. coli* BL21(DE3) and Δ*csg*D strains (* *p* < 0.05, ** *p* < 0.01 and *** *p* < 0.001).

**Table 1 ijms-24-13681-t001:** Summary statistics for the parent strain and Δ*csg*D strain based on RNA-seq data.

Sample	Total Reads	GC Content (%)	Q30 (%)	Mapped Reads	Mapped Ratio (%)
Parent strain-1	33,823,474	52.74	96.97	31,023,820	91.72
Parent strain-2	28,198,342	52.81	97.27	23,954,412	84.95
Parent strain-3	17,748,764	51.49	96.98	15,749,686	88.74
Δ*csg*D-1	25,372,190	53.08	93.61	23,002,214	90.66
Δ*csg*D-2	25,189,478	52.91	97.44	23,089,232	91.66
Δ*csg*D-3	28,254,794	52.84	96.83	26,430,290	93.54

**Table 2 ijms-24-13681-t002:** Partial differentially expressed genes in the *csg*D-deficient mutant strain.

Classification	Gene ID	Gene	Description
Biological adhesion	HO396_05355	*pga*A	partially deacetylated poly-*β*-1,6-N-acetyl-D-glucosamine export outer membrane porin PgaA
	HO396_09555	*mot*B	flagellar motor protein MotB
	HO396_02470	*fim*A	type 1 fimbrial major subunit FimA
	HO396_02475	*fim*C	type 1 fimbriae periplasmic chaperone FimC
Response to stimulus	HO396_01705	*ira*P	anti-adapter protein IraP
	HO396_05005	*omp*A	outer membrane protein OmpA
	HO396_07480	*osm*C	peroxiredoxin OsmC
	HO396_08475	*suf*E	cysteine desulfuration protein SufE
	HO396_11145	*ela*B	stress response protein ElaB

## Data Availability

Data is contained within the article or Appendix A. The more detailed data presented in this study are available on request from the corresponding author.

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
