# Peer review of "The Transcription Factor CsgD Contributes to Engineered Escherichia coli Resistance by Regulating Biofilm Formation and Stress Responses"

_ijms, 2023, doi:10.3390/ijms241813681_

Round 1
Reviewer 1 Report
In this article, the authors present the importance and contribution of the transcription factor CsgD in resistance of engineered Escherichia coli by regulating of biofilm formation and stress responses. This paper has a clear and logical structure structure and it is well written.
The most important points are discussed sufficiently and I find it very informing, so I would like to recommend it for the publication without any revisions.
Author Response
We greatly appreciate your excellent comments on this manuscript.
Reviewer 2 Report
Dear authors,
please respond to the following remarks:
· Please change "floating state" into "planktonic state".
· Subscription of figure one: If an alpha of 5 % was applied, statistical significance is indicated only by p<0.05! Differentiation into 0.01 or 0.001 is not a valid measure! Please correct.
· In 3.1. please provide information on how long bacteria were cultivated? How was the bacterial density? How was the bacterial concentration determined? Please provide additional information upon these issues!
· In 3.3. what was the initial bacterial concentration? Please provide more detailed information
· How many test batches were arranged? How many controls were used? Please provide detailed information about both issues!
· How the controls were arranged?
· In 3.7. How was the bacterial concentration of the seed solution? Please provide additional information.
English is fine, minor corrections are necessary.
Author Response
Response: We greatly appreciate your excellent comments on this manuscript. All comments were accepted. The full text has been revised by Dr. Richard Ansah Herman. In addition, English has been polished by professional institutions. The manuscript has been revised and explained carefully based on the comments. Please see the following responses.
- Please change "floating state" into "planktonic state".
Response: We have changed “floating state” to “planktonic state”. (Page 1, Line 42)
- Subscription of figure one: If an alpha of 5 % was applied, statistical significance is indicated only by p<0.05! Differentiation into 0.01 or 0.001 is not a valid measure! Please correct.
Response: We have changed “*p <0.05, ** p <0.01, and *** p <0.001” to “The asterisks wereobserved with statistically significant difference between the E. coli BL21(DE3) and ΔcsgD strains (p < 0.05).”. (Page 3, Line 116-117; Page 5, Line 183-184; Page 10, Line 315-316)
- In 3.1. please provide information on how long bacteria were cultivated? How was the bacterial density? How was the bacterial concentration determined? Please provide additional information upon these issues!
Response: The details of bacterial culture are described as follows: All strains with the concentration of 107 CFU/mL were inoculated at a 1% (v/v) ratio in Luria–Bertani (LB) medium. Strains were cultured at 37℃ for 12-14 h until the optical density at 600 nm (OD600 nm) of ~0.6, and the bacterial density was determined by gradient dilution method. (Page 10, Line 319-323)
- In 3.3. what was the initial bacterial concentration? Please provide more detailed information
Response: The description of bacterial concentration in the manuscript has been revised. The seed liquid of two strains with the concentration of 107 CFU/mL was inoculated in liquid LB mediumat a volume ratio of 1:100. (Page 11, Line 346-347, 352)
- How many test batches were arranged? How many controls were used? Please provide detailed information about both issues!
Response: In the determination of bacterial growth curve, three repeated experiments were set up for E. coli BL21(DE3) and ΔcsgD strains. The cell density of 3 biological replicates was measured and the average value was calculated, and a growth curve was plotted. LB medium of uninoculated strains served as a negative control. (Page 11, Line 348-349)
In the determination of biofilm content, two strains were inoculated in 96-well plates, and each group was set with 6 wells as parallel. In addition, the wells without inoculated strains were regarded as a negative control. (Page 11, Line 354-356)
- How the controls were arranged?
Response: In the determination of biofilm content, the strains were inoculated in a 96-well plate. The LB medium without inoculated strains was used as a negative control group, which could avoid the experimental error caused by the non-specific adsorption of crystal violet by 96-well plate. (Page 11, Line 354-356)
- In 3.7. How was the bacterial concentration of the seed solution? Please provide additional information.
Response: We have modified it to “Sterile glass slides were placed in a 12-well plate, inoculated with 1% bacterial seed solution with the concentration of 107 CFU/mL, and placed in an incubator at 37 °C for 24 h.” (Page 12, Line 15)
Reviewer 3 Report
In this study, the authors created a deletion mutant of the E. coli csgD regulatory gene using a CRISPR/cas9 approach and compared the parent strain with the mutant for a variety of traits including biofilm formation (including loosely and tightly bound, protein and polysaccharide content), motility, reactive oxygen species content, catalase production, and survival rates following exposure to acid, alkaline, oxidative, and osmotic stress. An RNAseq approach was used to examine differentially regulated expression of genes between the parent and mutant strains. Expression levels of representative genes identified were validated using a qPCR approach. The results obtained were largely consistent with the extensive literature on CsgD roles in the Gram-negative bacteria. The manuscript describes an impressive amount of work on the roles of CsgD in E. coli. However, there is a fundamental flaw in the experimental design that substantially limits the conclusions that can be drawn from the results of these experiments. The critical feature that is missing in the described studies is the inclusion of a genetically complemented strain. In the course of constructing the deletion mutant, unintended secondary mutations could have arisen and the observed phenotypes may result from these, rather than directly from the csgD deletion. Given the ease of genetic manipulation of E. coli, the lack of a complemented strain in these studies is a serious flaw.
Specific comments:
1. Lines 42-43: “In addition, biofilms are indispensable in the process of bacterial attachment and biofilm formation” is a poor sentence. First, biofilms are indispensable for biofilm formation is too obvious. Second, since this is a general statement, it ignores the myriad of examples of bacterial attachment that do not involve biofilm formation.
2. Lines 45-48: Need a reference here in support of this statement.
3. Line 49: All bacteria are equal in terms of heredity and phenotype.
4. Lines 53-55: You state that engineered bacteria are more vulnerable to environmental stresses and your cited reference is a paper in which bacteria were engineered to be MORE resistant to phenolic compounds. You need some actual evidence that this general statement is true (which I doubt is actually correct).
5. Lines 55-57: “biofilms play an important role in alleviating the survivability of E. coli in stressful environments”? Makes the bacteria more likely to die?
6. Line 105: Referring to OD595 immediately after discussing growth differences is confusing. You should state that Figure 1D shows the results of biofilm measurements.
7. Line 132: Since LB and TB are not standard abbreviations and this is the first instance of their usage, they should be defined as loosely and tightly bound, respectively.
8. Line 146: The morphology was complete? What would an incomplete morphology look like?
9. Lines 152-155: You need to express these results in more quantitative terms, rather than the general qualitative descriptors.
10. Figure 3D: The ND indicated above the delta csgD bar needs to be defined.
11. Line 221: down-regulated “directly or indirectly” by CsgD. This is especially important as your RNA seq data indicated differential expression of a variety of transcriptional regulators.
12. Line 235: galaxy metabolism?
13. Lines 271-273 and 286-288: The data show the down-regulation was not the trivial amounts stated in the text. Actually, the values were the residual expression levels, not the fold-down regulated.
14. Table S1: The BS21(DE3) strain has the genotype E. coli str. B F– ompT gal dcm lon hsdSB(rB–mB–) λ(DE3 [lacI lacUV5-T7p07 ind1 sam7 nin5]) [malB+]K-12(λS). It is the parent strain of the constructs used in this study, but it is NOT a wild-type E. coli strain.
15. Section 3.5: It is unclear how the different types of motility were distinguished and reference 44 provided no guidance on this.
Moderate issues as pointed out in the comments to the authors.
Author Response
Response: We greatly appreciate your excellent comments on this manuscript. The RNA-seq data of csgD deletion strain were separately aligned to the reference genome with Escherichia coli BL21(DE3) (CP053602.1), and no mutant genes except csgD gene were found. The csgD deletion strain obtained in this study was constructed by CRISPR/cas9 method. In this process, the donor DNA did not carry resistance genes, which minimized the interference of exogenous genes on the phenotype of the strain. Meanwhile, previous studies have shown that CsgD was a transcription factor in some gram-negative bacteria, especially pathogenic Escherichia coli, and plays an important role in the formation of biofilm, such as regulating the synthesis of fimbrial protein Curli. This study focused on the effect of CsgD on Escherichia coli in environmental stress. The results showed that the deletion of csgD gene led to the accumulation of reactive oxygen species in Escherichia coli, and its tolerance in acid, alkali, oxidation and osmotic stress environment was also significantly reduced. In addition, the results of RNA-seq and qPCR also showed that CsgD had a positive regulatory effect on stress-related genes, indicating that CsgD played a key role in the stress resistance of Escherichia coli. In conclusion, some regulatory functions of CsgD gene have been well known, and the current research results were enough to support our final conclusion. Therefore, it is not necessary to construct complemented strain in this study, and it would not make this study get additional conclusions. Thank you again for your valuable comments on this study. In addition, the full text has been revised by Dr. Richard Ansah Herman. The manuscript has been revised and explained carefully based on the comments. Please see the following responses.
- Lines 42-43: “In addition, biofilms are indispensable in the process of bacterial attachment and biofilm formation” is a poor sentence. First, biofilms are indispensable for biofilm formation is too obvious. Second, since this is a general statement, it ignores the myriad of examples of bacterial attachment that do not involve biofilm formation.
Response: What we want to express here is that biofilm is a group way that bacteria are ubiquitous in nature and resist unfavorable environment. But the specific description is not very appropriate, so we modify it to “Biofilm is not only a simple bacterial group, but also involves necessary physiological and phenotypic changes. It is a highly structured multicellular microbial community, which is composed of bacterial cells and extracellular polymeric substance (EPS) matrix, which could provide niches and enhance stress resistance” (Page 1-2, Line 42-45)
- Lines 45-48: Need a reference here in support of this statement.
Response: We added a reference (Engineered biofilm: innovative nextgen strategy for quality enhancement of fermented foods. Front Nutr. 2022, 9, 808630, doi:10.3389/fnut.2022.808630.) as support for biofilm applications. (Page 2, Line 48)
- Line 49: All bacteria are equal in terms of heredity and phenotype.
Response: It is obvious that Escherichia coli has a high degree of genetic and phenotypic diversity, and its genome is highly plastic and evolvable. There may be ambiguity in our statement here, so we modify it as follows:“Escherichia coli has been applied to bioengineering platform because of its clear genetic background and simple genetic engineering operation.” (Page 2, Line 49-50)
- Lines 53-55: You state that engineered bacteria are more vulnerable to environmental stresses and your cited reference is a paper in which bacteria were engineered to be MORE resistant to phenolic compounds. You need some actual evidence that this general statement is true (which I doubt is actually correct).
Response: The engineering strain was selected based on its advantages in one aspect, for example E. coli BL21(DE3), which has excellent protein expression ability. However, the stress resistance and biofilm formation ability of engineering strains were not the main factors to be considered in industrial application. Previous studies have shown that the biofilm formation ability of engineering strains (E. coli B and E. coli K12) was not ideal (Genome rearrangements induce biofilm formation in Escherichia coli C-an old model organism with a new application in biofilm research). Especially in the process of biological reaction, the growth state of engineering strains would be threatened by reaction conditions (acid, alkali and other organic compounds). Therefore, engineered E. coli was more vulnerable to multiple environmental stresses in application, and we added additional references to provide support for our views. (Page 2, Line 56)
- Lines 55-57: “biofilms play an important role in alleviating the survivability of E. coli in stressful environments”? Makes the bacteria more likely to die?
Response: We have revised it to “It is worth noting that biofilms play an important role in increasing the survivability of E. coli in stressful environments.” (Page 2, Line 56-57)
- Line 105: Referring to OD595 immediately after discussing growth differences is confusing. You should state that Figure 1D shows the results of biofilm measurements.
Response: This is an omission that should not appear, and we have revised it to “The determination results of biofilm (Figure 1d) show that the OD595nm value of ΔcsgD was only 0.66 ± 0.02, which was significantly lower than that of the wild-type strain (p < 0.05).” (Page 3, Line 121-123)
- Line 132: Since LB and TB are not standard abbreviations and this is the first instance of their usage, they should be defined as loosely and tightly bound, respectively.
Response: For abbreviations that first appear, we have added the full spelling. and modified it to “in loosely bound EPS (LB-EPS) and tightly bound EPS (TB-EPS)”. (Page 4, Line 134-135)
- Line 146: The morphology was complete? What would an incomplete morphology look like?
Response: The integrity of cell morphology shows that both two kinds of Escherichia coli are rod-shaped bacteria with blunt round ends and the cell structure has not changed. Therefore, the deletion of csgD gene would not cause morphological damage of Escherichia coli.
- Lines 152-155: You need to express these results in more quantitative terms, rather than the general qualitative descriptors.
Response: The fluorescence intensity of the stained biofilm was semi-quantitatively analyzed with reference to the method previously reported. The results showed that the biofilm fluorescence intensity of the wild-type strain was 7.01AU, which was significantly higher than ΔcsgD (2.15 AU). The content in the manuscript been modified to state that the results showed that “the fluorescence intensity of the wild-type strain (7.01 AU) was significantly higher More than that of ΔcsgD (2.15 AU)”. (Page 4, Line 154-156)
Reference: Quantitative analysis of histological staining and fluorescence using ImageJ. Anatomical Record-Advances in Integrative Anatomy and Evolutionary Biology. 2013, 296, 378-381. doi:10.1002/ar.22641
- Figure 3D: The ND indicated above the delta csgD bar needs to be defined.
Response: The size of “*” above ΔcsgD bar in Figure 3D has been adjusted. (Page 5, Line 177)
- Line 221: down-regulated “directly or indirectly” by CsgD. This is especially important as your RNA seq data indicated differential expression of a variety of transcriptional regulators.
Response: CsgD protein could be used as a transcription factor, which could be combined with the promoter of downstream genes to promote or inhibit the transcription process of downstream genes. In addition, CsgD could indirectly affect the formation of biofilm by affecting other signal transduction pathways or regulating intracellular metabolic processes. Therefore, to describe the role of CsgD in the biofilm formation and metabolism of bacterial more accurately, it is expressed as “CsgD could directly or indirectly regulate diverse metabolism-related genes”.
- Line 235: galaxy metabolism?
Response: We have changed “galaxy metabolism” to “galactose metabolism”. (Page 6, Line 238)
- Lines 271-273 and 286-288: The data show the down-regulation was not the trivial amounts stated in the text. Actually, the values were the residual expression levels, not the fold-down regulated.
Response: We have revised the description of the relative expression level of genes as required. The relative expression level of the gene of csgD-deficient strain was calculated by 2−ΔΔCt method, so the values were not the fold-down regulated. (Page 9, Line 275-277, 290-292)
- Table S1: The BS21(DE3) strain has the genotype E. coli str. B F– ompT gal dcm lon hsdSB(rB–mB–) λ(DE3 [lacI lacUV5-T7p07 ind1 sam7 nin5]) [malB+]K-12(λS). It is the parent strain of the constructs used in this study, but it is NOT a wild-type E. coli strain.
Response: The BL21(DE3) strain is the parent strain of csgD defective strain, so we modified “the wild-type strain” as “the parent strain” in Table S1 as required. (Supplement material, Table S1)
- Section 3.5: It is unclear how the different types of motility were distinguished and reference 44 provided no guidance on this.
Response: The measurement of bacterial motility is described in more detail. The strains were inoculated into LB semisolid medium containing 0.3%, 0.5% and 1.15% agar for swimming motility, swarming motility and twitching motility, respectively. The radius of the motion area was determined when strains were incubated at 37°C for 48 h. Meanwhile, a more detailed reference about bacterial motility was cited.
Reference: Swimming, swarming, twitching, and chemotactic responses of Cupriavidus metallidurans CH34 and Pseudomonas putida mt2 in the presence of cadmium. Arch Environ Con Tox. 2014, 66, 407-414, doi:10.1007/s00244-013-9966-5. (Page 11, Line 369-371)
Reviewer 4 Report
The original work entitled "The Transcription Factor CsgD Contributes to Engineered Escherichia coli Resistance by Regulating Biofilm Formation and Stress Responses" is properly described and contains scientifically significant research issues. The manuscript presents a number of different microbiological and genetic techniques to determine the involvement of the CsgD regulator in E. coli physiology.
To further enhance the quality of the manuscript, I would like to suggest a few modifications:
- Gene names everywhere should be written in italics (currently their spelling seems completely random)
- Line 44-45: Aggregates contribute to biofilm formation rather than the opposite; please modify accordingly
- Line 50: I believe that you wanted too express a term “shorter cell division” not “shorter growth cycle”
- Line 56: “alleviating the survivability” -> the very opposite (increasing the survivability)
- Line 61-62: “However, the presence of biofilms was not always optimistic.” -> the sentence does not sound very scientifically, please modify
- Lines 84-87: This sentence is too long, please split into two shorter
- Line 174-175: “weakened physiological processes such as the spread, colonization and infection of the strain” -> this was not proved scientifically here, please modify to “weakened physiological processes such as the spreading
- Figure 4 and line 235: Why in KEGG functional enrichment analysis "biofilm formation - Vibrio cholerae" was used instead of "biofilm formation - Escherichia coli"?
- Line 243: “cayalytic” -> catalytic
- Lines 316, 341, 382: Please indicate the bacterial cell density which was used (expressed preferably in CFU/mL)
- Line 359: Methodological inconsistency. The description says that Coomassie Brilliant Blue was used to assess the amount of proteins, while under Figure 1 it was written that it was the Lowry’s method
- In my opinion, the conclusions are too long and are a very faithful repetition of the abstract. Please shorten and present this passage as the most important thought from the current research.
The language used in the manuscript is correct and requires only minor corrections.
Author Response
Response: We greatly appreciate your excellent comments on this manuscript. All comments were accepted. The full text has been revised by Dr. Richard Ansah Herman. In addition, English has been polished by professional institutions. The manuscript has been revised and explained carefully based on the comments. Please see the following responses.
- Gene names everywhere should be written in italics (currently their spelling seems completely random)
Response: We have checked and modified all the italics in the manuscript, especially "csgD". (Line 20, 24, 26, 74, 96,161, 211, 222, 227-229)
- Line 44-45: Aggregates contribute to biofilm formation rather than the opposite; please modify accordingly
Response: What we want to express here is that the relationship between biofilms and population behavior of bacterial. But the specific description is not very appropriate, so we modify it to “It is a highly structured multicellular microbial community, which is composed of bacterial cells and extracellular polymeric substance (EPS) matrix, which could provide niches and enhance stress resistance” (Page 1-2, Line 42-45)
- Line 50: I believe that you wanted too express a term “shorter cell division” not “shorter growth cycle”
Response: We have changed “shorter cell division” to “shorter growth cycle”. (Page 2, Line 51)
- Line 56: “alleviating the survivability” -> the very opposite (increasing the survivability)
Response: We have changed “alleviating the survivability” to “increasing the survivability”. (Page 2, Line 56-57)
- Line 61-62: “However, the presence of biofilms was not always optimistic.” -> the sentence does not sound very scientifically, please modify
Response: Here, what we want to express is the duality of biofilm, which could provide protection for probiotics and pathogenic bacteria. We have revised it to “However, the protective effect of biofilm also brings trouble to the elimination of pathological bacteria”. (Page 2, Line 62-63)
- Lines 84-87: This sentence is too long, please split into two shorter
Response: We have revised it to “Firstly, the growth curve, biofilm formation ability and motility of bacteria were investi-gated. Then, by comparing the survival rates of wild-type strains and csgD-deficient strains in stress environment (acid stress, alkaline stress, oxidation stress and osmosic stress), the influence of csgD gene on the stress resistance of E. coli was evaluated”. (Page 2, Line 84-87)
- Line 174-175: “weakened physiological processes such as the spread, colonization and infection of the strain” -> this was not proved scientifically here, please modify to “weakened physiological processes such as the spreading
Response: We have revised it to “Therefore, the deletion of the csgD gene decreased the comprehensive motility of E. coli and weakened physiological processes such as the spreading” (Page 5, Line 175-176)
- Figure 4 and line 235: Why in KEGG functional enrichment analysis "biofilm formation - Vibrio cholerae" was used instead of "biofilm formation - Escherichia coli"?
Response: We have changed “biofilm formation - Vibrio cholerae” to “biofilm formation”. (Page 7-8, Figure 4c and Line 238-239)
- Line 243: “cayalytic” -> catalytic
Response: We have changed “cayalytic” to “catalytic”. (Page 8, Line 244)
- Lines 316, 341, 382: Please indicate the bacterial cell density which was used (expressed preferably in CFU/mL)
Response: The gradient dilution method was used to count the E. coli seed solution growing to logarithmic phase, and the bacterial cell density was about 107 CFU/mL. (Page 10-12, Line 319-320, 346-347, 352, 388-389)
- Line 359: Methodological inconsistency. The description says that Coomassie Brilliant Blue was used to assess the amount of proteins, while under Figure 1 it was written that it was the Lowry’s method
Response: Here is a mistake, we have changed “Lowry’s method” to “Coomassie Brilliant Blue G-250” in Figure 1. (Page 3, Line 113-114)
- In my opinion, the conclusions are too long and are a very faithful repetition of the abstract. Please shorten and present this passage as the most important thought from the current research.
Response: We have refined the conclusion to ensure its importance in the current research, rather than simply repeating the abstract.
Conclusions: Transcription activator CsgD was essential for the biological characteristics of engineered E. coli, including biofilm formation, motility, stress response and so on. CsgD could directly or indirectly regulate metabolism, cell membrane and catalytic activity. More importantly, CsgD has a positive regulatory effect on genes related to bacterial flagella synthesis, adhesion and stress response, and its abnormal expression would reduce the environmental stress resistance of engineered E. coli. These results reveals the molecular mechanism of CsgD in the stress resistance and provides potential targets for the treatment and control of E. coli. (Page 12, Line 422-429)
Round 2
Reviewer 2 Report
Dear authors,
thank you for correcting your manuscript. There are no further revisions necessary.
Author Response

(The authors gave the same response as above.)
